# Glutamatergic Neurons Differentiated from Embryonic Stem Cells: An Investigation of Differentiation and Associated Diseases

**DOI:** 10.3390/ijms22094592

**Published:** 2021-04-27

**Authors:** Jen-Hua Chuang, Wen-Chin Yang, Yenshou Lin

**Affiliations:** 1School of Life Science, National Taiwan Normal University, Taipei 116, Taiwan; chuangjenhua5@gmail.com; 2Agricultural Biotechnology Research Center, Academia Sinica, Taipei 115, Taiwan; wcyang@gate.sinica.edu.tw

**Keywords:** glutamatergic neurons, embryonic stem cells, differentiation, neurodegeneration, Alzheimer’s disease

## Abstract

Neurons that have been derived from various types of stem cells have recently undergone significant study due to their potential for use in various aspects of biomedicine. In particular, glutamatergic neurons differentiated from embryonic stem cells (ESCs) potentially have many applications in both basic research and regenerative medicine. This review summarized the literatures published thus far and focused on two areas related to these applications. Firstly, these neurons can be used to investigate neuronal signal transduction during differentiation and this means that the genes/proteins/markers involved in this process can be identified. In this way, the dynamic spatial and temporal changes associated with neuronal morphology can be investigated relatively easily. Such an in vitro system can also be used to study how neurons during neurogenesis integrate into normal tissue. At the same time, the integration, regulation and functions of extracellular matrix secretion, various molecular interactions, various ion channels, the neuronal microenvironment, etc., can be easily traced. Secondly, the disease-related aspects of ESC-derived glutamatergic neurons can also be studied and then applied therapeutically. In the future, greater efforts are needed to explore how ESC-differentiated glutamatergic neurons can be used as a neuronal model for the study of Alzheimer’s disease (AD) mechanistically, to identify possible therapeutic strategies for treating AD, including tissue replacement, and to screen for drugs that can be used to treat AD patients. With all of the modern technology that is available, translational medicine should begin to benefit patients soon.

## 1. Introduction

Glutamatergic neurons, which use glutamate as their neurotransmitter, form the majority of excitatory neurons and occupy a major part of the brain cortex. Based on histological findings, these glutamatergic neurons also seem to be crucially involved in a range of diseases, including Alzheimer’s disease (AD), since one of the characteristics of the pathophysiological state of such diseases is the atrophy and/or death of neurons in the cerebral cortex/hippocampus. It is clear that an understanding of the molecular mechanisms that underlie the differentiation of various stem cells into glutamatergic neurons is likely to be useful to the above. This could lead to the development of cellular models of these diseases, which in turn would allow the investigation of their pathophysiological mechanisms. Such findings would then allow the exploration of new therapeutic strategies, and the creation of better approaches to the screening of drugs to treat these diseases. Even more importantly, engineered functional cells might have the potential to be used in cellular therapy and allow the replacement of impaired neurons and tissue. However, there are several different types of stem cells which include embryonic stem cells (ESCs), inducible pluripotent stem cells (iPSCs), adult mesenchymal stem cells, and umbilical stem cells, all of which might be used as a source of such neurons. Thus far, ESCs, iPSCs, and adult neuronal stem cells have been shown to be capable of differentiating into neurons. Each one of these has its own advantages and disadvantages in terms of their applications and purposes. For example, genomic instability needs to be concerned more on iPSCs than on ESCs. Nevertheless, iPSCs are easier to obtain than ESCs and adult neuronal stem cells in terms of ethics and availability. Furthermore, ESCs and iPSCs have more potential to differentiate into variety of cell types than adult neuronal stem cells. When differentiated neurons are ready for transplantation, the resources of using iPSCs and adult neuronal stem cells seem to be less reject than those using ESCs. However, the issue on ESCs might be circumvented by somatic cell nuclear transfer technique which will be discussed later. Therefore, based on the genetic stability and differentiation potency, the ESC-derived cellular differentiation is worth exploring its biology and using as a choice/alternative of translational medicine. It is worth noting that several crucial protocols were developed in the mid-2000’s so as to differentiate ESCs and/or iPSCs into neurons/glutamatergic neurons with high purity [1,2]. This review focuses on the use of mouse ESCs (mESCs) and human ESCs (hESCs)-derived neurons to produce, in particular, glutamatergic neurons, and targets two specific aspects, firstly, basic research, and secondly, clinical applications.

## 2. As Part of Differentiation Research

As a zygote develops into the stage of blastocyst in vivo, this embryo can be collected by surgery before implanting in the uterus. The inner cell mass (ICM) within the blastocyst was isolated and cultivated in the appropriate medium to obtain so-called hESCs/mESCs. Upon ligands/stimuli to initiate the differentiation, these cells are capable of developing into ectoderm, mesoderm, or endoderm cells in vitro. In general, this review looks at five characteristics of the differentiation process and these are schematically summarized in Figure 1.

### 2.1. The Study of Stimuli and Signal Transduction during Development/Differentiation

Development/differentiation processes are known to include survival, growth, and morphological changes. Before differentiation, ESCs must maintain themselves in a condition that allows them to propagate normally, while at the same time keeping their ability to self-renew. Many previous reviews have described these steps clearly and thoroughly [3,4,5]. Upon differentiation, signal transduction occurs; this involves the relaying of stimuli as a command that is then transferred via the relevant signaling molecules. Chuang et al. has comprehensively reviewed the possible stimuli and signaling systems involved in differentiation in vitro [6]. The inducers and pathways explored include retinoic acid, transforming growth factor/bone morphogenetic protein, fibroblast growth factor, cytokine, Hedgehog, Notch, Wnt/b-catenin, JAK, STAT, c-Jun N-terminal kinase/mitogen-activated protein kinase, SMAD and others. Notwithstanding this, we now include some up-to-date information concerning this area in this review. Specifically, a transcriptome analysis was performed to investigate the roles of retinoic acid receptor (RAR), retinoid X receptors (RXR), and liver X receptor (LXR) target genes in retinoic acid (RA)-induced neurogenesis in mESCs [7]. The comprehensive data obtained from this study offer a great resource for gaining further understanding of cell fate specification. Furthermore, after coculturing mESCs and trophoblast stem cells (TSCs) in vitro, Junyent et al. recently found that mESCs are able to generate cytonemes, which are signaling filopodia that function as recruiting signals in response to TSCs-secreted Wnts. This Wnt signaling, along with other events, eventually results in embryogenesis [8].

### 2.2. The Identification of the Genes, Proteins and Markers Involved in the Cellular Differentiation Process

It has been reported that the differentiation of ESCs into neurons in vitro recapitulates the developmental processes in vivo both in general [9,10,11] and in particular anatomical areas, such as the hippocampus [12]. For example, utilizing primary cortical neurons and/or brain tissue culture, homeobox gene *T cell leukemia 3* (*Tlx3*), also known as *HOX11L2*, has been found to promote the differentiation of glutamatergic neurons in the central nervous system. In agreement with previous in vivo data, both *Tlx3* and *Pax6*, have been verified to be involved in the differentiating of mESCs in vitro [13,14]. More molecules observed during ESCs differentiating into glutamatergic neurons in vitro include the following. The glutamatergic neuronal subtype promoted by *Tlx3* has been found to be attributable to direct binding between a chromatin modifier CBP (cyclic adenosine monophosphate (cAMP)-response element binding protein (CREB) binding protein) and the Tlx3 homeodomain [15]. TATA-binding protein-associated factor 4 (TAF4), a subunit in TFIID, has been shown to play roles in embryogenesis, such as the correct patterning of trunk and anterior structures in vivo. As supporting evidence, during this study mESCs were constructed with a *TAF4a* gene knockout to examine its importance. The *TAF4a^−/−^* mESCs failed to differentiate into either glutamatergic neurons or cardiomyocytes in vitro [16]. Brn2, a homeodomain transcription factor also known as Pou3f2, has been found to play an essential role in the neurogenesis of sub-ventricular zone progenitor cells and cortical layer pyramidal neurons in vivo [17,18]. By utilizing ChIP-seq and small hairpin RNA, Urban et al. were further able to show that the transcription factor Brn2–Zic1 axis is important for the neuronal differentiation induced by retinoic acid in vitro [19]. Uda et al. found that the activation of P2Y4, a nucleotide receptor, in mESCs, is able to increase the differentiation of glutamatergic neurons that are characterized by the presence of the vesicular glutamate transporter (vGLUT) marker [20]. The ectopic expression of transcription factor neurogenin 2 (Ngn2) alone is sufficient to induce the differentiation of mature glutamatergic neurons [21]. The analysis of gene expression during this process has further revealed the interaction partners and target genes during the in vitro differentiation induced by Ngn2 partially resemble neurogenesis in vivo. Utilizing a reverse engineering approach, a gene named *E130012A19Rik* (*E13*) seems to be one of the master regulators of gene expression in mESCs that are undergoing differentiation into glutamatergic neurons in vitro [22]. This study also confirmed that this gene is expressed during E13 in the cerebral cortex of adult mice, as well as during mouse cerebral cortex development. Interestingly, miRNAs, such as miR-200 family, miR-429, miR-302 family, and miR-17-92 cluster members, were inferred to play crucial roles in the process of ES/neurons differentiation [23]. The function of the individual gene/protein/marker is briefly summarized in Table 1.

The aforementioned genes/proteins/markers represent the individual cases from various studies of the differentiation process. Nonetheless, the relationships between and among them needs further investigation. A much more comprehensive approach using paired-end RNA sequencing was used to investigate the transcriptional changes during the differentiation of mESC-derived glutamatergic neurons [24]. The summarized expression profiles can be accessed at the website http://dx.doi.org/10.6084/m9.figshare.154969. This dataset provides very valuable information on differentiation during the glutamatergic development of ESCs in vitro. Therefore, it is quite clear that a comparison of the genes and proteins involved in the differentiation of glutamatergic neurons in vivo and in vitro shows many important similarities.

### 2.3. The Use of 3-Dimensional (3D) Methods to Investigate Dynamic Spatial and Temporal Changes during Neuronal Morphology

Changes in morphology, such as neurite formation, have become some of the critical issues when studying differentiation because such changes are related to the functioning of neurons. Morphological change is traditionally investigated using normal 2-dimensional (2D) culture methods. However, using 3-dimensional (3D) methods have provided a novel way of investigating neuron formation and differentiation. In this context, it is worth noting that a number of different protocols have been used to differentiate ESCs in 3D, and other types of stem cell such as iPSCs, into glutamatergic neurons. Based on the differentiation procedure used, these are able to be divided into two categories. One approach involves scaffold-based structures, such as using 3D templates, [25,26,27] while the other approach involves scaffold-free systems, such as spheroids, embryoid bodies, and organoids [28,29,30,31]. Such approaches are considered to more precisely replicate differentiation in vivo and as a result, these 3D approaches have gained more attention when trying to investigate the details of the process in a more natural environment. Nevertheless, the use of various stem cell systems other than ESCs has seen significant progress. For example, one of the systems involved the seeding of human neural stem cells (hNSCs) onto synthetic multifunctionalized hydrogels. The results from these novel 3D studies have shown that such approaches are able to overcome some of the limitations associated with previous 3D culture systems, such as the biomaterial used and the composition of the culture medium. During the process, differentiated cells are able to form entangled networks that contain functional and mature neural phenotypes [32,33]. Yan et al. have characterized the effects of biomolecule-directed differential responses during 3D cultures by utilizing hiPSCs [34]. Using a rotational mixer at low speed, followed by agitation, static culture, and agitation again, Collins and Haigh also developed a protocol for neuronal culture that produced adult murine neural stem cells in 3D; this has allowed the investigation of cellular functioning in vitro [35]. Finally, using ESCs derived neurons, Dubois-Dauphin et al. were able to cultivate mESC-differentiated neurons on a hydrophilic polytetrafluoroethylene (PTFE) membrane and create a 3D model. Furthermore, they were able to characterize the long-term survival of the differentiated neurons by examining their morphology, their electrophysiological activity, and their neurotransmitter release [36]. Therefore, the glutamatergic neurons that have been differentiated from ESCs provide an excellent model for observing dynamic spatial and temporal processes. When the ESC-derived glutamatergic neurons were cultivated in 3D circumstance, there are methods to investigate the spatio–temporal changes of differentiating neurons. For example, the digital approach is adopted to dynamically observe the neuronal morphology [37,38]. This definitely further broadens our knowledge on neuronal/synaptic growth and migration, synaptic integration, signal transmission, network connectivity and circuit dynamics, etc.

### 2.4. Regulation and Functions of Extracellular Matrix (ECM) Secretion, Ion Channels, the Neuronal Microenvironment, among Others

The connections between signaling, genes, cytoskeletal remodeling, and the extracellular matrix (ECM) eventually reflect the objectives of neuronal differentiation. It has been found that N-cadherin, a synaptic adhesion molecule, is able to regulate short-term plasticity at glutamatergic synapses. When N-cadherin is absent from glutamatergic neurons differentiated from mESCs, short-term plasticity is greatly altered and synaptic depression is enhanced [39]. Furthermore, neural cell adhesion molecule (NCAM) is able to promote the differentiation of hippocampal precursor cells, which were isolated on embryonic day 16.5, into a neuronal lineage cells, specifically glutamatergic neurons [40]. Although their system uses stem cells other than ESCs, Aizman et al. also found that the ECM plays a crucial aspect in neuropoiesis [41]. When ion channels are examined, a study using glutamatergic neurons differentiated from hESCs found that both sodium and calcium channel subunits were expressed and a delayed rectifier potassium channel current was also exhibited [42]. Interestingly, the neurospheroids involved in neural microenvironment during differentiation are attracting more attention. Simao et al. used neurospheroids to address microenvironment remodeling during the neural differentiation of human stem cells [43]. The result demonstrated that proteome, transcriptome, and synaptic expression showed significant changes at cell membrane, ECM, and basement membrane during 3D differentiation as compared to 2D differentiation. Cai et al. were able to model neurodegenerative diseases such as AD by using the acoustofluidic method to rapidly construct 3D neurospheroids and inflammatory microenvironments [44]. Utilizing embedded 3D bioprinting, Li et al. also develop a 3D brain-like co-culture structure for studying the cell–cell interactions, neurospheroids/glia and the brain microenvironment [45]. Nonetheless, the effect of the neuronal microenvironment on ESC-derived glutamatergic neurons, and vice versa, both require more investigation and pinpointing of some groundbreaking findings.

### 2.5. Neurons That Have Been Developed Using Neurogenesis In Vitro Are Able to Integrate into Existing Tissue

A number of studies have focused on how various types of neurons develop and can then be integrated in vivo, as covered in the relevant literature over several decades. For example, there are studies on the migration of glutamatergic neurons and on cortical patterning during development; these have been reviewed in detail elsewhere [46]. Recently, it was shown that pluripotent stem cell-derived neurons can integrate adult host neural networks also in a human-to-human grafting situation [47]. To date, how progenitor cells and neurons that have been derived from the various other types of stem cells in vitro are able to eventually successfully integrate into existing tissue remains a major issue. Thus, this area continues to be challenging, with differentiation over time being explored using 3D models or at the organism level. Some reports, using ESCs, as well as other stem cells-derived neurons, have been promising. Wernig et al. found that the functional integration of ESC-derived neurons does occur in vivo [48]. Specifically, they found that neurons derived from ESCs, but expressing GFP as a cell marker, are able to be implanted into the cerebral ventricles of embryonic rats within a variety of brain regions. These neurons acquire complex morphologies and seem to adopt neurotransmitter phenotypes. In another study, Benninger et al. reported that the functional integration of mESC-derived neurons into hippocampal slice cultures was able to be detected [49]. In addition, Chen et al. found that cultured subventricular zone progenitor cells transduced with neurogenin-2 were to become mature glutamatergic neurons and integrate into the dentate gyrus [50]. Co-cultivating mESC-derived neurons with neocortical explants, Copi et al. found that those neurons exhibited some electrophysiological parameters that are similar to those of primary cultured neocortical neurons. They also found that activity and BDNF were able to induce plasticity within the miniature synaptic currents in mESC-derived neurons and that these cells could be integrated into a neocortical network [51]. The above results indicate that ESC-derived glutamatergic neurons in an in vitro system are suitable for facilitating in vivo studies, even when it is relatively difficult to trace how differentiated neurons integrate existing brain tissue.

## 3. In Pathophysiological Studies

The majority of neurons that form the brain cortex and hippocampus are glutamatergic neurons. Hence, using ESCs differentiated into glutamatergic neurons is an obvious approach when exploring disease-related questions. In a study of the potential neurotoxic effects of silver nanoparticles, Begum et al. used hESC-derived glutamatergic neurons as the experimental system and found that citrate-coated silver nanoparticles were able to induce glutamate excitotoxicity by mediating signaling molecules such as that increased in the phosphorylation of glycogen synthase kinase-3 α/β at Tyr216 and of Tau at Ser396 [52]. In a similar way, Wang et al. recently investigated the effect of chronic exposure to bisphenol-A on glutamatergic neurons that had been derived from hESCs. They found that bisphenol-A-induced neurotoxicity seemed to result from an increased production of reactive nitrogen species (RNS) and reactive oxygen species (ROS) [53]. The application of ESC-derived neurons has also been used in many other research areas, for example, when investigating auditory neurons. These studies differentiated hESCs into neural precursor cells (NPCs) and cultivated them on aligned nanofiber mats. These NPCs ultimately differentiated into glutamatergic neurons, as well as producing in vitro neurites projections along the nanofibers; these neurons might be expected to be useful when transplanted to replace auditory nerves [54,55]. In another pathophysiological context, Lin et al. studied the effect of amyloid-β (Aβ) on hESC-derived cortical neurons. These found that AMPAR-mediated whole-cell current and excitatory postsynaptic current were significant decreased after treatment with Aβ. They further showed the reduction in currents and glutamatergic synaptic transmission were able to be restored by inhibitors of histone methyltransferase [56]. Utilizing the modification of one allele at the normal APP locus to directly express a secretory form of Aβ40 or Aβ42, Ubina et al. attempted to establish a cellular model for hESC-derived neurons and used this to investigate neurodegeneration. Whole transcriptome RNA-Seq analysis and pathway/annotation analysis suggest that such an approach using such a cellular model for the study of AD is feasible [57]. Hence, the possible applications of hESC-derived glutamatergic neurons to AD-related research areas have gained significant attention and progress.

## 4. Future Perspectives

### 4.1. A Potential Therapeutic Strategy for the Treatment of AD via Neurons/Tissues Replacement

One of the choices to treat AD patients might be just to replace the damaged neurons/tissues with functional ones in specifically affected brain areas. There are several options in terms of generating the regeneration of neurons/tissues for such an approach. One such source is ESC-derived neurons. By taking advantage of somatic cell nuclear transfer (SCNT), patient-matched glutamatergic neurons derived from ESCs could be used to circumvent the rejection problem associated with transplantation [58]. Utilizing the SCNT technique in association with various differentiation methods, the ESCs could be differentiated into the desired cell types and then used to treat patients. This approach has been described as “therapeutic cloning” [59]. The recent improvement in 3D culture and in the SCNT technique means that it is possible to envision using ESC-derived glutamatergic neurons for the treatment of a range of neurodegenerative diseases, particularly AD (Figure 2).

### 4.2. Screening Drugs for the Treatment of AD

The disease of AD is characterized as atrophy/impairment in the hippocampus and brain cortex area where the majority of neurons is the glutamatergic neurons. Typically, drug screening to treat diseases such as AD uses a number of different strategies; these are generally based on a series of molecular targets or changes at a cellular level. When it comes to cellular level testing and/or high content screening, most studies use neurons derived from a cell line, a primary culture, iPSCs, or ESCs. Although cell line-derived neurons, and even a cell line alone, are highly convenient and less costly, their characteristics do not resemble the reality within the patient’s body closely, particularly when compared to other sources of cells. In this context, primary glutamatergic neuronal cells seem to provide the most informative results because they are directly derived from a living organism such as a rodent. However, such an approach means that many animals need to be sacrificed in order to keep the high content screen working at full capacity, and this approach evidently cannot be applied to the case of obtaining human primary neurons. Currently, neurons differentiated from iPSCs that have been obtained from patients are most likely to reflect most closely the pathophysiological status of that individual. Nevertheless, glutamatergic neurons derived from ESCs, along with the addition of beta amyloid, are likely to provide an alternative for investigating molecular mechanisms and carrying out high content drug screening when targeting a disease such as AD. That is, the use of ESCs might provide a better genomic stability and the well-established protocols to differentiate glutamatergic neurons. Chuang et al. previously developed an approach to consistently obtain a homogeneous population of glutamatergic neurons differentiated from mESCs [60,61]. These neurons are a response to the addition of potassium chloride (KCl) (Figure 3). Thus, these neurons and the relevant platform might provide a great opportunity to screen for effective compounds from a wide variety of sources, which will allow them to be developed as treatments for AD. It is also clear that ESC-derived glutamatergic neurons are likely to be useful as part of personalized medicine if the aforementioned SCNT technique is used in association with therapeutic cloning [58,62].

## 5. Conclusions

ESC-derived glutamatergic neurons provide a valuable source of materials for differentiation research and translational medicine. In the field of development/differentiation, signal transduction during differential processes can be easily investigated using these neurons. In this way, signaling molecules are able to be identified when a stimulus is given to the cells. The genes/proteins/markers that change during the signaling event, and lead to the differentiation process, are then easy to pinpoint and/or target. Biological functioning is reflecting on the dynamic spatial and temporal changes that affect neuronal morphology and these can be explored in detail during such experiments. Using these neurons, it is possible to investigate the mechanism by which neurons develop during in vitro neurogenesis. Furthermore, this will also allow us to see how such neurons integrate existing normal brain tissue. The major factors involved in this process include extracellular matrix (ECM) secretion, various interaction between molecules, changes in ion channels, and the neuronal microenvironment. Experiments using ESC-derived glutamatergic neurons are a convenient way of exploring the regulation and function within these cells.

From the clinical research point of view, there are far fewer reports on the applications of ESC-derived glutamatergic neurons. Therefore, it is very important that, in the future, efforts in this area should be marshaled in a way that allows the development of new therapeutic strategies and/or the creation of new high throughput drug screening approaches. In particular, these efforts should target neurodegenerative diseases such as AD and other similar diseases. When developing a drug screening system and therapeutic approaches, neurons differentiated from ESCs, after modification using the SCNT technique, are likely to be suitable for therapeutic cloning and personalized medicine. As research continues, ESC-derived glutamatergic neurons should be able to solve some of the mysteries associated with neuron biology. These approaches should also help move new ideas from the bench to the bedside via translational medicine. Such innovations are likely to especially benefit AD patients.

## Figures and Tables

**Figure 1 ijms-22-04592-f001:**
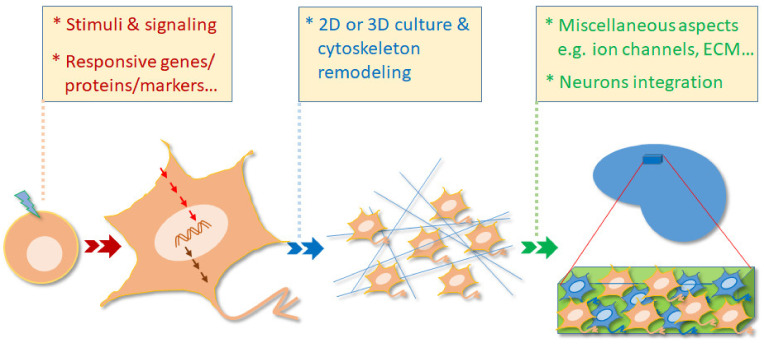
The ESC-derived glutamatergic neurons provide a great model for the study of differentiation. During differentiation, various stimuli, a range of relevant signaling molecules, and various responsive genes/protein/markers are able to be conveniently investigated in vitro. Furthermore, these glutamatergic neurons are also able to be cultivated in either a 2D environment or in a 3D matrix in order to study cytoskeleton remodeling and morphological changes. When these neurons are grafted into brain tissue, neuronal integration, ion channel expression, extracellular matrix secretion, as well as many other aspects, are able to be explored from a functional point of view.

**Figure 2 ijms-22-04592-f002:**
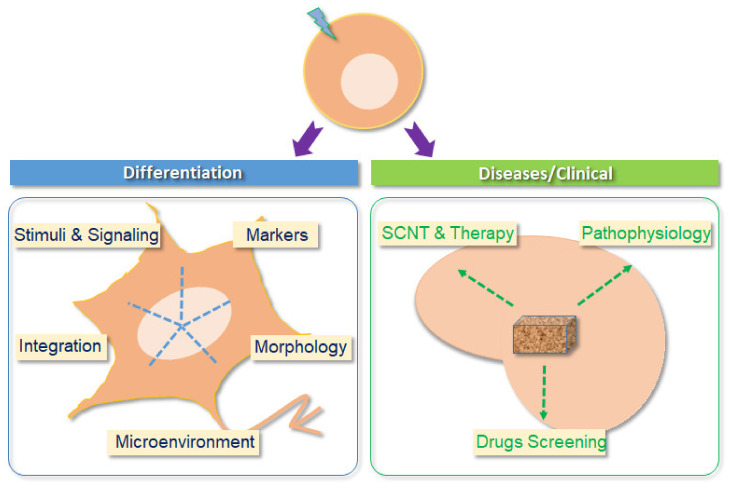
Two major aspects of the application of glutamatergic neurons have been differentiated from ESCs. One is to investigate the differentiation process. Both structure and function can be conveniently studied in this context. The other aspect, which is more difficult, is to develop personalized and precision medicine approaches so that this eventually becomes a practical approach to treating patients. Furthermore, using glutamatergic neurons derived from human and/or mouse ESCs, disease mechanisms and high throughput/content drug screening can be easily performed in vitro. Thus, glutamatergic neurons might one day be used as replacements to repair impaired neurons/tissues in patients. SCNT, somatic cell nuclear transfer.

**Figure 3 ijms-22-04592-f003:**
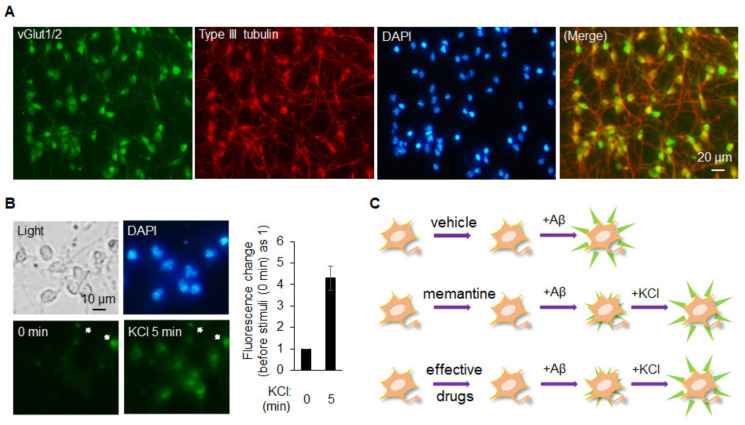
Glutamatergic neurons differentiated from ESCs could provide an alternative resource for establishing the drug screening system/platform: (**A**) neurons differentiated from mESCs for around 7 days were fixed and immunocytochemistry was performed using vGLUT1/2 or type III tubulin as the primary antibody. A merged image of vGLUT1/2 (green) and type III tubulin (red) is shown; (**B**) neurons differentiated from mESCs were incubated with 1 µM DiBAC4(3), a slow-response voltage sensitive fluorescent dye, for 30 min in KRH solution prior to the experiment. The fluorescent intensity was monitored and recorded every minute after 45 mM KCl was added to the solution. Only the images before (0 min) and after (5 min) the stimulation are shown here. The arrows indicate the nonspecific signals which are used as controls for fluorescent intensity. Intensity changes were quantified, plotted and are shown on the right panel. Data are shown as means ± SD; (**C**) a cartoon is drawn schematically to indicate the use of these neurons for drug screening platform. Top panel, Aβ_1-42_-induced abnormal depolarization will be observed. Middle panel, memantine, a drug used as part of the clinical treatment of AD patients, shall be able to block the Aβ_1-42_-induced abnormal depolarization. At the end of experiment, these neurons shall be stimulated with KCl to confirm their responsiveness. The use of memantine functions as a positive control for the platform. Bottom panel, the effective drugs can be screened by using the same logic with the one in memantine.

**Table 1 ijms-22-04592-t001:** The genes, proteins and markers involved in the cellular differentiation process in vitro.

Genes/Proteins/Markers	Functions
*Tlx3* (*HOX11L2*)	Promotes the differentiation of glutamatergic neurons in the central nervous system [13,15].
*Pax6*	Plays a very early role in the specification of retinal neurons and in the developing cortex [14].
TAF4	Is a subunit in TFIID and has been shown to play roles in embryogenesis [16].
Brn2 (Pou3f2)	Brn2 and Brn2–Zic1 axis are important in neuronal differentiation induced by retinoic acid [17,18,19].
P2Y4	Extracellular nucleotides can mediate this nucleotide receptor to induce glutamatergic markers [20].
Ngn2	Is a helix–loop–helix TF and is sufficient to induce the differentiation of ESCs into glutamatergic neurons [21].
*E30012A19Rik*	One of the master regulators of gene expression for ESCs to differentiate into glutamatergic neurons [22].
miRNAs	miR-200 family and miR-17-92 cluster members are relevant in the neuronal differentiation of ESCs [23].

*Tlx3*, T-cell leukemia homeobox protein 3; *Pax 6*, Paired box 6; TAF4, TATA-binding protein-associated factor 4; Brn2/Pou3f2, pou domain class 3 transcription factor 2; Ngn2, Neurogenin 2.

## Data Availability

Not applicable.

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
