# Peer review of "Glutamatergic Neurons Differentiated from Embryonic Stem Cells: An Investigation of Differentiation and Associated Diseases"

_ijms, 2021, doi:10.3390/ijms22094592_

Round 1
Reviewer 1 Report
In this paper authors propose an interesting review of the current literature about the use of Embryonic Stem Cells (ESCs) as a source of functional glutamatergic neurons to study differentiation pathways and associated diseases, with a particular focus on Alzheimer’s Disease (AD).
I think this review should be helpful for scientist in the field, but some aspects should be better addressed.
Following some suggestions:
1.General observation: it is not clear for the reader the real advantage for the usage of this system respect to other types of stem cells.
Line 42-50; 273-275: the authors state that there are different types of stem cells that could be used as a source of neurons, but every cell type mentioned has his own advantages and disadvantages. There are no mentions about what these advantages and disadvantages are.
This statement should be better explained with particular focus on what are the advantages of ESCs compared to other types.
2.On the use of ESCs in AD.
Lines 273-280: as in point 1, a real advantage for the use of these models compared to existing ones, that are nowadays broadly used for the study of many aspects of this disease, did not emerge.
As in 1, a better elaboration with focus on AD should be provided.
Figures.
In general, the quality of figure is low, maybe is just the format of uploaded version, if not, improve resolution.
Fig. 1-2: It is not clear if these are original figures, if not, refer in the legends to original papers and licences/permissions if needed.
Fig.1 some genes are reported in the figure and relative pathways explained in the text. A table that resumes the function and the involvement in neuronal differentiation of these genes would be helpful for the reader.
Fig. 3. refer in the legends to original paper and licences/permissions if needed.
Legend for fig 3.A “Neurons differentiated from mESCs from around 7 days were recorded using light microscope”, it is not clear what is the exact meaning of the term “recorded”, is there an electrophysiological trace that has not been included in the figure? Also, in the text in lines 279-280 it is stated that neurons in figure 3 are responsive to KCl and beta amyloid, I would expect an electrophysiological trace showing the depolarization induced by KCl or, instead, an explanation of what type of responsiveness is mentioned (also for beta amyloid). Check or correct these points.
Author Response
Response to Reviewers
Manuscript: ijms-1150394
Title: Glutamatergic neurons differentiated from embryonic stem cells: an investigation of differentiation and associated diseases
___________________________________________________________________________
Response to Reviewer #1
1.”General observation: it is not clear for the reader the real advantage for the usage of this system respect to other types of stem cells.
Line 42-50; 273-275: the authors state that there are different types of stem cells that could be used as a source of neurons, but every cell type mentioned has his own advantages and disadvantages. There are no mentions about what these advantages and disadvantages are.
This statement should be better explained with particular focus on what are the advantages of ESCs compared to other types.”
We now add a few sentences to describe the advantages and disadvantages of various stem cells. We also emphasize the use and advantage of ESCs compared to other types. Many thanks to the reviewer for pointing this out.
“For example, genomic instability needs to be concerned more in iPSCs than in ESCs. Nevertheless, iPSCs is easier to obtain than ESCs and adult neuronal stem cells in terms of ethics and availability. Furthermore, ESCs and iPSCs have more potential to differentiate into variety of cell types than adult neuronal stem cells. When differentiated neurons are ready for transplantation, the resources of using iPSCs and adult neuronal stem cells seem to be less rejection issue than using ESCs. However, the issue on ESCs might be circumvented by somatic cell nuclear transfer technique which will be discussed later. Therefore, based on the genetic stability and differentiation potency, the ESCs-derived cellular differentiation is worth exploring its biology and using as a choice/alternative of translational medicine.” These sentences were added into line 48-57 in the revised manuscript.
- “On the use of ESCs in AD.
Lines 273-280: as in point 1, a real advantage for the use of these models compared to existing ones, that are nowadays broadly used for the study of many aspects of this disease, did not emerge.
As in 1, a better elaboration with focus on AD should be provided.”
Indeed, iPSCs-derived neurons have been widely used as the started material to screen drugs against related diseases including AD. Recently, mesenchymal stem cells are also used to test the therapeutic effect by administering into mice brain/body in AD mice model. However, the promising effect seems to attribute to via many factors such as exosomes but not via neurons (Guo et al., 2020 Alzheimer's Res. Ther. 12:109). While we searched the keywords “ESCs”, “glutamatergic”, and “drug screening” on PubMed, two literatures pop up. None of these two is reporting any study on drug screening. We recently found that the response after Abeta stimulation on ESCs-derived glutamatergic neurons seem to be comparable to the one seen on primary glutamatergic neurons (unpublished data). Therefore, we propose that these ESCs-derived glutamatergic neurons might also be a great alternative resource for drug screening in AD in addition to using iPSCs-derived neurons.
Since we are discussing the differentiated neurons used for drug screening in this paragraph (line 262 in original manuscript), we now add a few sentences to address reviewer’s concern regarding of the focus on AD and the use of ESCs in AD. “The disease of AD is characterized as atrophy/impairment in the hippocampus and brain cortex area where the majority of neurons is the glutamatergic neurons.” This sentence is added into line 324-325 in revised manuscript. “That is, the use of ESCs might provide a better genomic stability and the well-established protocols to differentiate into glutamatergic neurons.” This sentence was added into line 341-343 in the revised manuscript.
Figures.
- “In general, the quality of figure is low, maybe is just the format of uploaded version, if not, improve resolution.”
We do apologize about the low resolution in jpg format regarding of these figures. Now we have improved the resolution by uploading/posting them with tiff files format.
- “Fig. 1-2: It is not clear if these are original figures, if not, refer in the legends to original papers and licences/permissions if needed.”
Once again, we do apologize about the low resolution in jpg format regarding of these figures. Now we have improved the resolution by uploading/posting them with tiff files format.
- “Fig.1 some genes are reported in the figure and relative pathways explained in the text. A table that resumes the function and the involvement in neuronal differentiation of these genes would be helpful for the reader.”
We take the reviewer’s great suggestion. A table that briefly states the function and the involvement in neuronal differentiation of these genes now is included.
- “Fig. 3. refer in the legends to original paper and licences/permissions if needed.
Legend for fig 3.A “Neurons differentiated from mESCs from around 7 days were recorded using light microscope”, it is not clear what is the exact meaning of the term “recorded”, is there an electrophysiological trace that has not been included in the figure?”
Fig 3A shows an image of neurons 7 days after mESCs differentiation under light microscope. Initially, we would like to simply present evidence for these mESCs-derived glutamatergic neurons through morphological/biochemical/cell biological methods in this review. These result is done separately and recently although we had published similar images in 2013. Actually, the three panels (panel A, B and C) in Fig 3 are all different from the published ones. As the review’s concern on license/permissions issues, we decide to leave Fig 3A and 3B out. Therefore, there is no problem on the regards of the term “recorded” because the description will be deleted.
“Also, in the text in lines 279-280 it is stated that neurons in figure 3 are responsive to KCl and beta amyloid, I would expect an electrophysiological trace showing the depolarization induced by KCl or, instead, an explanation of what type of responsiveness is mentioned (also for beta amyloid). Check or correct these points.”
Because the number of compounds that may have potential to treat AD is overwhelming, the speed and efficacy of a drug screening platform are critical issues. Many laboratories and we use DiBAC4(3), a fluorescent dye sensitive to changes in membrane potential, to monitor the KCl-induced and/or Abeta-induced depolarization. Therefore, the depolarization seen in the responsiveness of neurons after KCl stimulation will reflect on the increased intensity of green fluorescence. We now explain these changes in the figure legend. At the same time, we also quantitate the result and show it beside the images. Many thanks to the reviewer for pointing this out. In addition, since the Abeta-induced depolarization on these mESCs-derived glutamatergic neurons might be a great alternative platform for drug screening, we draw schematically a cartoon to present it as a future perspective as Fig 3C.
during 3D differentiation as comparing to 2D differentiation. Cai et al were able to model neurodegenerative diseases such as AD by using coustofluidic method to rapidly construct 3D neurospheroids and inflammatory microenvironments (Cai et al, 2020 Analyst 145(19):6243-6253). Utilizing embedded 3D bioprinting, Li et al also develop a 3D brain-like co-culture structure for studying the cell-cell interactions, neurospheroids/glia and brain microenvironment (Li et al 2020 Biofabrication).” These sentences were added to line 228-238 in the revised manuscript. The references number are #43, #44 and #45.
- “Description of the methods involved in performing the experiments in figure 3 missing.”
The methods and materials of experiments in figure 3 are revised and described now. These are shown in line 353-366 in the revised manuscript.
- “Images displayed in figure3C-D could be displayed with higher resolution. Authors should be able to plot the changes in intensity rather than just show them in the images.”
We do apologize about the low resolution in jpg format regarding of these figures. Now we have improved the resolution by uploading/posting figure 3C and 3D with tiff files format.
In addition, we have quantitated the result of changes in intensity and shown beside the images. Many thanks to the reviewer for pointing this out. Because one of the reviewers concerns about the license/permissions issues, we decide to leave the original Fig 3A and 3B out but keep Fig 3C and 3D. Now they are renamed to Fig 3A and 3B. We draw schematically a cartoon to present that Abeta-induced depolarization on these mESCs-derived glutamatergic neurons are a great alternative platform for drugs screening. This is shown as Fig 3C.
Minor comments:
- “Figure 1 appears pixelated. Labels in the figure such as A, B and C helps in better description of the figure. If not, it could be divided by faint lines to provide clear distinction.”
Once again, we do apologize about the low resolution in jpg format regarding of these figures. Now we have improved the resolution by uploading/posting figures with tiff files format. As the reviewer’s great suggestion, we decided to separate the area by faint lines/boxes in the figure so as to provide clear distinction.
- “Some of the text in figure legends is in blue – Ngn2, Neurogenin2”
Many thanks to the reviewer for point it out. The text color in blue in the figure legend is revised.
- “The colours in the figure 2 make it hard to read the text, please reconsider the colours.”
We take the reviewer’s great suggestion. The color in the figure 2 is now redraw to make it become much more reader-friendly.
Reviewer 2 Report
Generation of neurons from embryonic stem cells (ESCs) have physiological relevance and various applications in regenerative medicine. The review from Chuang, et al focuses on these neurons with relevance to signal transduction and several players involved in the process. Further it points on how this is relevant in extracellular matrix secretion, molecular interactions, ion channels, neuronal microenvironment on one side. On the other side the review is focused on therapeutic relevance of ESC-derived glutamatergic neurons. The review is very important since glutamatergic neurons regulate cognitive, emotion, and motor functions and dysfunction glutamatergic neurons can not only lead to neurological diseases like Alzheimer’s disease as pointed out in review but also Down syndrome. However, there are some things which can be addressed to make this more comprehensive:
Lack of references in the introduction. Authors fail to acknowledge the work from DV Schaffer, Yan Liu, M Bibel, etc.
Authors should focus a bit on origin of ESCs which provides a molecular foundation for differentiation.
Signalling pathways such as JAK-STAT3, MEK/ERK, PKC, SMAD have to be discussed since biomedical applications of ESCs depend on these signalling pathways involved and their downstream effectors.
While discussing the methods to investigate spatio-temporal changes authors could discuss about digital approaches, gain and loss of function approaches, etc.
Discussion regarding neurosperoids involved in neural microenvironment during differentiation and how they influence the expression of cellular and extracellular features in the tissue is missing.
Description of the methods involved in performing the experiments in figure 3 missing.
Images displayed in figure3C-D could be displayed with higher resolution. Authors should be able to plot the changes in intensity rather than just show them in the images.
Minor comments:
Figure 1 appears pixelated. Labels in the figure such as A, B and C helps in better description of the figure. If not, it could be divided by faint lines to provide clear distinction.
Some of the text in figure legends is in blue – Ngn2, Neurogenin2
The colours in the figure 2 make it hard to read the text, please reconsider the colours.
Author Response
Response to Reviewers
Manuscript: ijms-1150394
Title: Glutamatergic neurons differentiated from embryonic stem cells: an investigation of differentiation and associated diseases
___________________________________________________________________________
Response to Reviewer #2
- “Lack of references in the introduction. Authors fail to acknowledge the work from DV Schaffer, Yan Liu, M Bibel, etc.”
As the review’s suggestion, we now add some description to acknowledge the work from different groups. So, “It is worth noting that several crucial protocols were developed in the mid-2000s so as to differentiate ESCs and/or iPSCs into neurons/glutamatergic neurons with high purity (Bibel et al, 2004 Nat Neurosci 7:1003-9 and Robertson et al, 2008 Front Biosci 3:21-51).” The sentence was added to line 57-59 in the revised manuscript. The references number are #1 and #2.
There are many key protocols to differentiate into neuronal types other than glutamatergic neurons such as the one for GABAergic neurons developed by Liu et al, 2013 Nat Protoc 8:1670-9. We did not include it due to the focus of this concise review. However, the very important work contributed by Liu’s group regarding the 3D culture of glutamatergic neurons (Chen et al, 2019 Acta Pharm Sin B 9:557-564) was now cited in the content. It is added to line 185 in the revised manuscript as a reference #27.
- “Authors should focus a bit on origin of ESCs which provides a molecular foundation for differentiation.”
We added a few sentences to describe the origin of ESCs which provides a molecular foundation for differentiation. “As a zygote develops into the stage of blastocyst in vivo, this embryo can be collected by surgery before implanting in the uterus. The inner cell mass (ICM) within the blastocyst was isolated and cultivated in the appropriate medium to obtain so called hESCs/mESCs. Upon ligands/stimuli to initiate the differentiation, these cells are capable to develop into ectoderm, mesoderm, or endoderm cells in vitro.” The sentence was added to line 64-68.
What the reviewer means might also be the focus on what cause ESCs to differentiation. In this aspect, we previous review the stimuli and signaling regarding of the ligands and signaling pathway on directing ESCs differentiate into glutamatergic neurons in vitro (Chuang et al, 2015 World J Stem Cells 7:437-47). In order not to repeat the same content and save space, we simply describe it as line 86-90 in the revised manuscript. The sentence is as follows. “The inducers and pathways explored include retinoic acid, transforming growth factor/bone morphogenetic protein, fibroblast growth factor, cytokine, Hedgehog, Notch, Wnt/b-catenin, JAK, STAT, c-Jun N-terminal kinase/mitogen-activated protein kinase, SMAD and others.”
- “Signalling pathways such as JAK-STAT3, MEK/ERK, PKC, SMAD have to be discussed since biomedical applications of ESCs depend on these signalling pathways involved and their downstream effectors.”
Indeed, we totally agree the review’s point. As the abovementioned rational and the published article, we simply describe it as line 86-90 in the revised manuscript.
- “While discussing the methods to investigate spatio-temporal changes authors could discuss about digital approaches, gain and loss of function approaches, etc.”
Many thanks to the reviewer’s reminder. Now we simply added a few sentences to describe the digital approaches but not other more approaches which might need a more comprehensive way to discuss them. “When the ESCs-derived glutamatergic neurons were cultivated in 3D circumstance, there are methods to investigate spatio-temporal changes of differentiating neurons. For example, the digital approach is adopted to dynamically observe the neuronal morphology (review Halavi et al., 2012 Frontier Neuroscience 6:1-11; Parekh et al., 2013 Neuron 77:1017-1038). This definitely further broaden our knowledge on neuronal/synaptic growth and migration, synaptic integration, signal transmission, network connectivity, and circuit dynamics etc.” These sentences were added to line 208-213 in revised manuscript. The references number are #37 and #38.
- “Discussion regarding neurosperoids involved in neural microenvironment during differentiation and how they influence the expression of cellular and extracellular features in the tissue is missing.”
Initially, we are wondering whether or not the niches surrounding the ESCs/neurospheroids and/or ESCs-differentiated glutamatergic neurons have been investigated in vitro. While searching PubMed using keywords “ESCs”, “glutamatergic”, and “neuronal microenvironment”, there is no literature shown. While using keywords “neurospheroids” and “neuronal microenvironment” on PubMed, there are four articles. Having digested the relevant articles, we simply added a few sentences to describe them. “Interestingly, the neurospheroids involved in neural microenvironment during differentiation is getting more attention. Simao et al used neurospheroids to address microenvironment remodeling during neural differentiation of human stem cells (Simao et al 2018 Stem Cell Reports 11:552-564). The result demonstrated that proteome, transcriptome, and synaptic expression showed significant changes at cell membrane, ECM, and basement membrane during 3D differentiation as comparing to 2D differentiation. Cai et al were able to model neurodegenerative diseases such as AD by using coustofluidic method to rapidly construct 3D neurospheroids and inflammatory microenvironments (Cai et al, 2020 Analyst 145(19):6243-6253). Utilizing embedded 3D bioprinting, Li et al also develop a 3D brain-like co-culture structure for studying the cell-cell interactions, neurospheroids/glia and brain microenvironment (Li et al 2020 Biofabrication).” These sentences were added to line 228-238 in the revised manuscript. The references number are #43, #44 and #45.
- “Description of the methods involved in performing the experiments in figure 3 missing.”
The methods and materials of experiments in figure 3 are revised and described now. These are shown in line 353-366 in the revised manuscript.
- “Images displayed in figure3C-D could be displayed with higher resolution. Authors should be able to plot the changes in intensity rather than just show them in the images.”
We do apologize about the low resolution in jpg format regarding of these figures. Now we have improved the resolution by uploading/posting figure 3C and 3D with tiff files format.
In addition, we have quantitated the result of changes in intensity and shown beside the images. Many thanks to the reviewer for pointing this out. Because one of the reviewers concerns about the license/permissions issues, we decide to leave the original Fig 3A and 3B out but keep Fig 3C and 3D. Now they are renamed to Fig 3A and 3B. We draw schematically a cartoon to present that Abeta-induced depolarization on these mESCs-derived glutamatergic neurons are a great alternative platform for drugs screening. This is shown as Fig 3C.
Minor comments:
- “Figure 1 appears pixelated. Labels in the figure such as A, B and C helps in better description of the figure. If not, it could be divided by faint lines to provide clear distinction.”
Once again, we do apologize about the low resolution in jpg format regarding of these figures. Now we have improved the resolution by uploading/posting figures with tiff files format. As the reviewer’s great suggestion, we decided to separate the area by faint lines/boxes in the figure so as to provide clear distinction.
- “Some of the text in figure legends is in blue – Ngn2, Neurogenin2”
Many thanks to the reviewer for point it out. The text color in blue in the figure legend is revised.
- “The colours in the figure 2 make it hard to read the text, please reconsider the colours.”
We take the reviewer’s great suggestion. The color in the figure 2 is now redraw to make it become much more reader-friendly.
Round 2
Reviewer 1 Report
I have no more comments on the revised version, all the concerns were addressed.
I recommend this review for publication